# Active vision of bees in a simple pattern discrimination task

HaDi MaBouDi[1,2,3]*, Jasmin Richter[1], Marie-Geneviève Guiraud[1,4], Mark Roper[1,5], James AR Marshall[2], Lars Chittka[1]

[1]School of Biological and Behavioural Sciences, Queen Mary University of London, London, United Kingdom; [2]Department of Computer Science, University of Sheffield, Sheffield, United Kingdom; [3]School of Biosciences, University of Sheffield, Sheffield, United Kingdom; [4]School of Natural Sciences, Macquarie University, Sydney, Australia; [5]Drone Development Lab, Ben Thorns Ltd, Colchester, United Kingdom

**Abstract** Active vision, the sensory-motor process through which animals dynamically adjust visual input to sample and prioritise relevant information via photoreceptors, eyes, head, and body movements, is well-documented across species. In small-brained animals like insects, where parallel processing may be limited, active vision for sequential acquisition of visual features might be even more important. We investigated how bumblebees use active vision to distinguish between two visual patterns: a multiplication sign and its 45°-rotated variant, a plus sign. By allowing bees to freely inspect patterns, we analysed their flight paths, inspection times, velocities and regions of focus through high-speed videography. We observed that bees tended to inspect only a small region of each pattern, with a preference for lower and left-side sections, before accurately accepting target or rejecting distractor patterns. The specific pattern areas scanned differed between the plus and multiplication signs, yet flight behaviour remained consistent and specific to each pattern, regardless of whether the pattern was rewarding or punishing. Transfer tests showed that bees could generalise their pattern recognition to partial cues, maintaining scanning strategies and selective attention to learned regions. These findings highlight active vision as a crucial aspect of bumblebees' visual processing, where selective scanning behaviours during flight enhance discrimination accuracy and enable efficient environmental analysis and visual encoding.

*For correspondence:
maboudi@gmail.com

## Editor's evaluation

This useful study revisits a challenging visual learning task in bumblebees. By analyzing the flight trajectory and body orientation in a learning paradigm, the authors discovered an active vision strategy. This contributes to our knowledge of the bumblebee flight behavior. The convincing work will be of interest to researchers working in neuroethology.

## Introduction

Active vision refers to a closed loop of sensory-motor process wherein animals actively manipulate their visual input, optimising perception and decision-making. In some vertebrates, the repertoire of such active vision strategies is well researched (*Land, 1999*; *Land and Nilsson, 2012*; *Yarbus, 2013*). To scan visual targets, there can be large-scale movement by the body or head, or smaller scale movements of the eyes (saccades) or photoreceptors (*Juusola and Song, 2017*; *Najemnik and Geisler, 2005*; *Yang and Chiao, 2016*) as well as subtle head movements or head-bobbing, which induce motion parallax to assess distances (*Dutta et al., 2020*; *Dutta and Maor, 2007*; *Kral, 2003*; *Kral, 1998*; *Sobel, 1990*). Such active vision is essential for obtaining an accurate three-dimensional representation of the

surrounding environment (*Kagan, 2012*; *Kemppainen et al., 2022*; *Martinez-Conde et al., 2013*; *Martinez-Conde and Macknik, 2008*; *Werner et al., 2016*). In certain vertebrates, eye movements are also used as a sampling strategy, enhancing the encoding of high spatial frequency of natural stimuli and providing fine spatial information (*Anderson et al., 2020*; ; *Kuang et al., 2012*; *Rucci and Poletti, 2015*). Some animals adopt characteristic routes during visual tasks to facilitate target recognition (*Skorupski and Chittka, 2017*; *Dawkins and Woodington, 2000*; *Langridge et al., 2021*; *Lehrer et al., 1985*). For instance, individual chickens exhibit stereotyped approach paths when learning to discriminate visual patterns, consistently using the same path and side relative to the object during each trial (*Dawkins and Woodington, 2000*). However, these paths are not identical across all chickens, as each bird develops its own distinct approach strategy. Interestingly, they fail at these tasks when their developed route is disrupted. Additionally, characteristic head movements have been observed in pigeons during image stabilisation for forward locomotion (*Theunissen and Troje, 2017*).

In insects, which possess miniature brains and, thus possibly more limited parallel processing, there appears to be a heightened necessity for sequential scanning to acquire spatial information compared to animals with larger brains (*Chittka and Niven, 2009*; *Chittka and Skorupski, 2011*; *Juusola et al., 2025*; *MaBouDi et al., 2020a*; *Spaethe et al., 2006*). Indeed, in bumblebees, there is evidence that complex patterns cannot be discriminated when they are only briefly flashed on a screen, preventing bees from sampling in a continuous scan (*Nityananda et al., 2014*). *Langridge et al., 2021* explored how bumblebees' approach direction prior to landing affects their colour learning, showing that bees tend to remember the colour they predominantly face during their approach, which influences their learned preferences for floral patterns. Furthermore, bees exhibit defined sequences of body movements in response to particular visual stimuli (*Collett et al., 1993*; *Guiraud et al., 2018*; *Lehrer, 1994*; *MaBouDi et al., 2020a*; *MaBouDi et al., 2020b*; *Werner et al., 2016*). This deterministic relationship between movement and perception suggests that active vision allows movement to shape the sensory information acquired and influences perception.

Extensive studies have demonstrated that bees are capable of memorising and discriminating a wide variety of visual patterns, including complex ones that, for example, include different stripe orientations in each of four quadrants (*Avarguès-Weber et al., 2011*; *Benard et al., 2006*; *Dyer et al., 2005*; *Guiraud et al., 2025a*; *Srinivasan, 2010*; *Srinivasan, 1994*; *Stach et al., 2004*; *Turner, 1911*; *Frisch, 1914*; *Wehner, 1967*). On the other hand, there is a long history of claims that bees are incapable of discriminating some relatively simple patterns (*Avarguès-Weber et al., 2011*; *Guiraud et al., 2022*; *Hertz, 1935*; *Hertz, 1929*; *Horridge, 1996*; *Srinivasan, 1994*; *Frisch, 1914*). As one example, it was reported that honeybees (*Apis mellifera*) were not able to distinguish a "plus pattern", made up of a vertical and horizontal bar, from the same pattern rotated through 45° that is multiplication sign pattern (*Horridge, 1996*; *Srinivasan et al., 1994*) in a Y-maze setup where the patterns were displayed at a fixed distance from the bees' decision point. However, there is evidence that the successes and failures of bees in discriminating visual patterns are not strictly related to pattern complexity. Factors such as the angle subtended on the visual field of a stimulus from a distance (*Horridge, 1996*; *Srinivasan, 2010*) and the particular sensitivity of orientation and colour selective neurons in the bee visual system (*Paulk et al., 2009*; *Paulk et al., 2008*; *Roper et al., 2017*; *Maddess and Yang, 1997*) may limit the ability of bees to solve a specific task. Indeed, the visual scanning procedures that bees use when examining and memorising the patterns can also affect performance (*Giurfa et al., 1999*; *Guiraud et al., 2025b*; *Lehrer, 1994*; *MaBouDi et al., 2023*; *Stach and Giurfa, 2005*).

In this study, we revisit one of the pattern discrimination tasks that has been proven highly challenging for honeybees (*Srinivasan et al., 1994*), namely the plus versus multiplication sign discrimination task (these experiments required bees to make a decision from a distance). Here, we aimed to explore whether and, more importantly, how bumblebees can successfully solve this discrimination task. Hence, by recording the bees' flight trajectories, and analysing their scanning movements, we aimed to determine the strategies employed in solving this visual task, specifically to investigate whether they are able to develop an active sampling strategy.

## Results

The main objective of this study was to examine whether bees develop specific scanning behaviours that enable them to solve pattern discrimination tasks, effectively. We conducted a visual discrimination

task using a plus sign and a 45°-rotated version of the same pattern (multiplication sign). Forty bumblebees (*Bombus terrestris audax*) from six colonies were utilised and housed in wooden nest boxes connected to a flight arena (*Figure 1A*). Two groups of bees (n=20 each) were trained using a differential conditioning protocol: one group was trained to associate the plus pattern with a reward (sucrose solution) while avoiding the multiplication pattern associated with punishment (quinine solution), and the other group was trained on the reciprocal arrangement (*Figure 1B*). After five bouts of training or upon reaching 80% performance in the last 10 choices, the bees underwent one learning test and three transfer tests.

The learning test used the same multiplication and plus patterns as during training to assess whether the bees developed specific manoeuvres for inspecting patterns before making choices. Each of the transfer tests presented novel stimuli showing fragments of both training patterns. In the bottom-half test, only the bottom half of the multiplication and plus patterns was shown, while in the top-half test, only the top half was presented. These tests aimed to assess whether the bees' recognition was based solely on the top or bottom halves of the patterns. Additionally, a single-bar orientation test was conducted to determine the bees' preferences for each component bar of the training stimuli (*Figure 1B*).

In all four tests, correct and incorrect stimuli were randomly positioned on the rear arena wall, with feeding tubes containing sterilised water (*Figure 1A*). The bees' flight trajectories during the tests were recorded using two high-speed cameras, capturing their flight behaviour from different angles. Advanced video analysis was then performed, using a custom algorithm developed to automatically detect and track the bees' movements. The algorithms extracted various behavioural parameters, including flight speeds, distances from the patterns, flight orientations, inspection times, and areas of interest (see Materials and methods section).

## Bumblebees' performance in a visual recognition task

We first confirmed that bumblebees, when allowed to fly as close to the patterns as they desired, could perform the simple visual discrimination task of identifying and associating either a plus or a multiplication sign pattern with a reward (sucrose solution) and the other with quinine solution (penalty, bitter taste).

Bees in Group 1 recognised the plus patterns as rewarding above chance after just 20 choices (*Figure 1C*; Wilcoxon signed rank test; z=1.95, n=20, p=0.04; mean = 57%). Conversely, Group 2 achieved the same performance on the multiplication patterns after 30 choices (Wilcoxon signed rank test; z=2.34 n=20, p=3.6e-3). Nevertheless, there was no notable difference in the learning rate between the two groups after 30 choices (p=0.77), and the bees' performance was not affected by colony membership (p=0.25; *Figure 1C*; see *Table 1*). The bees continued to increase in performance during the 70 choices of the training (per block of 10 choices, see *Figure 1C*); whereupon all bees achieved ≥92% (±7.8 SEM) correct choice performance. The results of a generalised linear mixed model (GLMM) analysis confirmed that both groups of bees had learned to select the rewarding patterns significantly above chance (>50%) after training (*Figure 1C*, p=4.8e-9). Additionally, the bees' performance in the learning tests indicated that both groups of trained bees successfully learned to discriminate the plus from the multiplication sign, and vice versa (*Figure 1D*; Wilcoxon signed rank test; z=3.21, n=13, p=1.3e-3 for Group 1; z=3.08, n=12, p=2.0e-3 for Group 2); again there was no significant difference between the performance of two groups in the learning test (Wilcoxon rank sum test; z=0.45, n=25, p=0.64). The bees' performance in the learning test was similar to that seen during the last block of 10 choices of the training phase (Wilcoxon signed rank test z=1.09, n=25, p=0.27).

During the learning tests, for 8 out of the 13 bees in Group 1 (trained to plus) the first pattern to be inspected (flight within 12 cm diameter of centre of pattern and 10 cm out from rear stimulus wall) was the rewarding plus pattern (*Figure 1E*). However, 5 out of the 12 bees in Group 2 also inspected one of the plus patterns first, in their case the incorrect pattern. Therefore, as a whole, the bees' pattern selection from a distance (i.e. from arena entrance to stimuli wall) was no different to chance (50% correct initial pattern inspections; *Figure 1E*; $\chi^2$ test, Chi-square statistics = *0.35*, p=0.55). In addition, during all the correct initial inspections of both groups, the bees still scanned the pattern before flying to the feeder tube, further suggesting that the bees do not receive sufficient evidence for a choice from a distance (*Figure 1—figure supplement 1*).

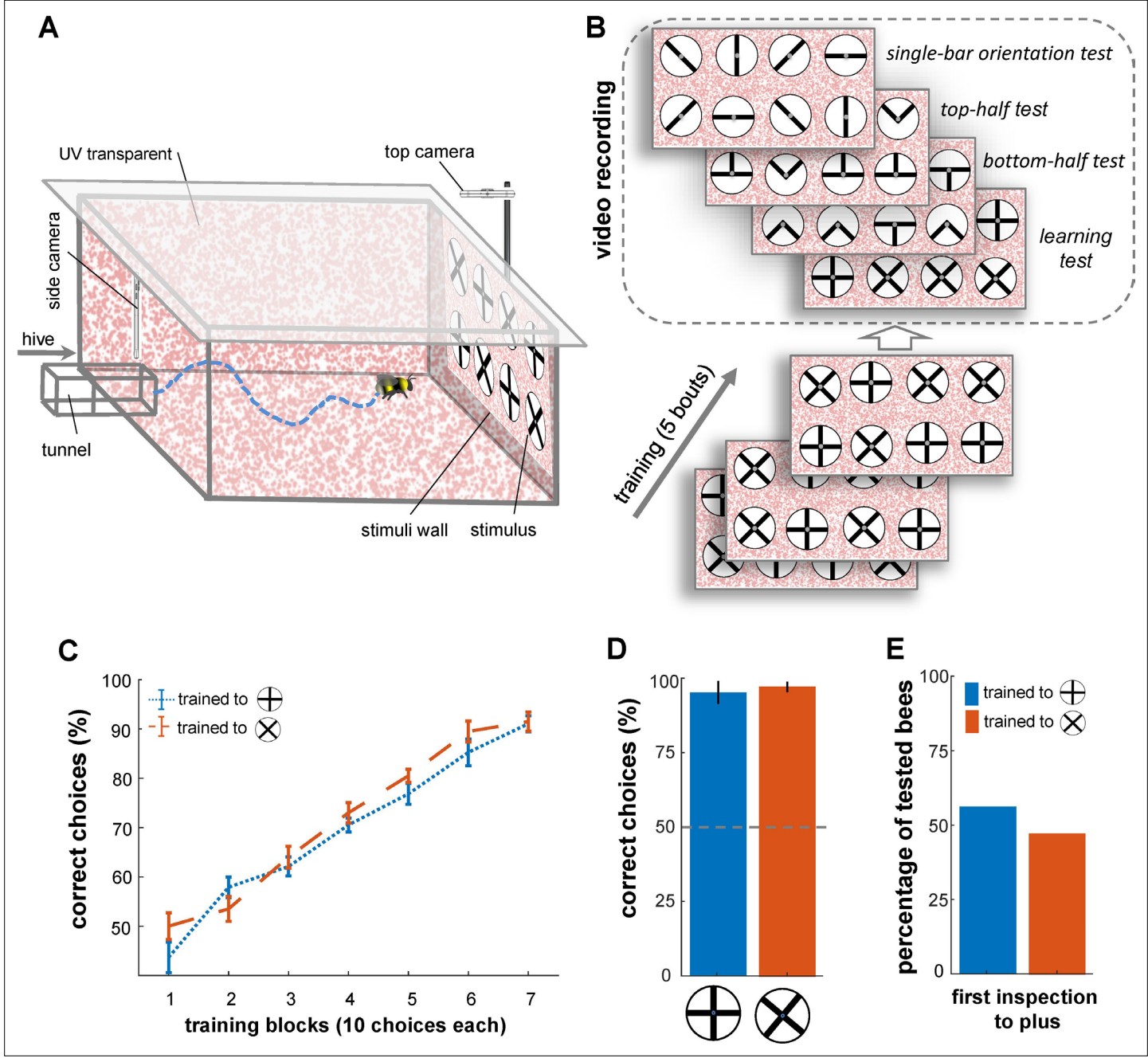

**Figure 1.** Bees' performance in a pattern recognition task. (**A**) Illustration of the flight arena, equipped with two cameras: a side camera above the entrance viewing the rear wall and a top camera above the rear wall recording from above. (**B**) Training and testing protocol: Bees were trained with four plus and four multiplication patterns on white discs (10 cm diameter, 2 mm margin), attached to the arena's rear wall *via* small Eppendorf tubes. Group 1 of bumblebees (n=20) was rewarded with the plus sign (sucrose solution) and punished with the multiplication sign (quinine), while Group 2 (n=20) received the opposite reinforcement. After training, four tests with refresher interval bouts were conducted: (1) a learning test using the original patterns from training, (2) a top-half test showing only the upper half of the patterns, (3) a bottom-half test displaying only the lower half of the patterns, and (4) a single-bar test presenting individual bars in different orientations. (**C**) Learning curves for Group 1 (blue) and Group 2 (orange) show both groups learned the pattern discrimination task. (**D**) Unrewarded learning test performance, with both groups distinguishing patterns (p<4.8e-3). (**E**) Initial inspections of the plus symbol upon arena entry showed no significant preference from a distance (>5 cm), with 8 out of 13 bees in group 1 and 5 out of 12 bees in group 2 making initial visits to the plus symbol in the learning test (p=0.55).

The online version of this article includes the following figure supplement(s) for figure 1:

**Figure supplement 1.** Bees' flight paths upon first entering the arena during learning tests.

**Table 1.** Summary of the Generalised Linear Mixed Model (GLMM) examining factors in relation to proportion of correct choices during the training.

The dependent variable was the number of correct choices from the block of 10 choices. Fixed factors, Colony, Group and Trial were examined in the model. Bee index was calculated in the model as a random factor. Model fit statistics: AIC = 32.91; BIC = 337.62; Log-Likelihood=−156.45; Deviance = 312.91.

| Fixed factors | Estimate | SE | tStat | DF | p Value | Lower | Upper |
|---|---|---|---|---|---|---|---|
| *Intercept* | −0.11 | 0.50 | −0.33 | 266 | 0.71 | −1.17 | 0.83 |
| *Colony* | 0.19 | 0.17 | 1.04 | 266 | 0.25 | −0.13 | 0.42 |
| *Group* | −0.06 | 0.27 | −0.24 | 266 | 0.82 | −0.61 | 0.36 |
| *Trials* | 0.21 | 0.03 | 6.75 | 266 | **4.8e-9** | 0.14 | 0.27 |

## Bees' discrimination of partial and component visual cues

To control for the possibility that the partial cues of patterns may have also influenced the bees' decisions, we carried out three transfer tests (see Methods section). *Figure 2* illustrates bumblebees' performance in the transfer testing phase, where they were presented with specific sections or components of the trained visual patterns (*Figure 1B*). In the bottom-half test, bees demonstrated a statistically significant ability to discriminate between the bottom portions of the correct and incorrect patterns (*Figure 2A*; Wilcoxon signed rank test; z=2.81, n=10, p=4.9e-3 for Group 1; z=2.66, n=10, p=7.7e-3 for Group 2), suggesting that they could generalise their learned associations to these bottom cues. In contrast, the top-half test revealed that bees exposed only to the upper sections of

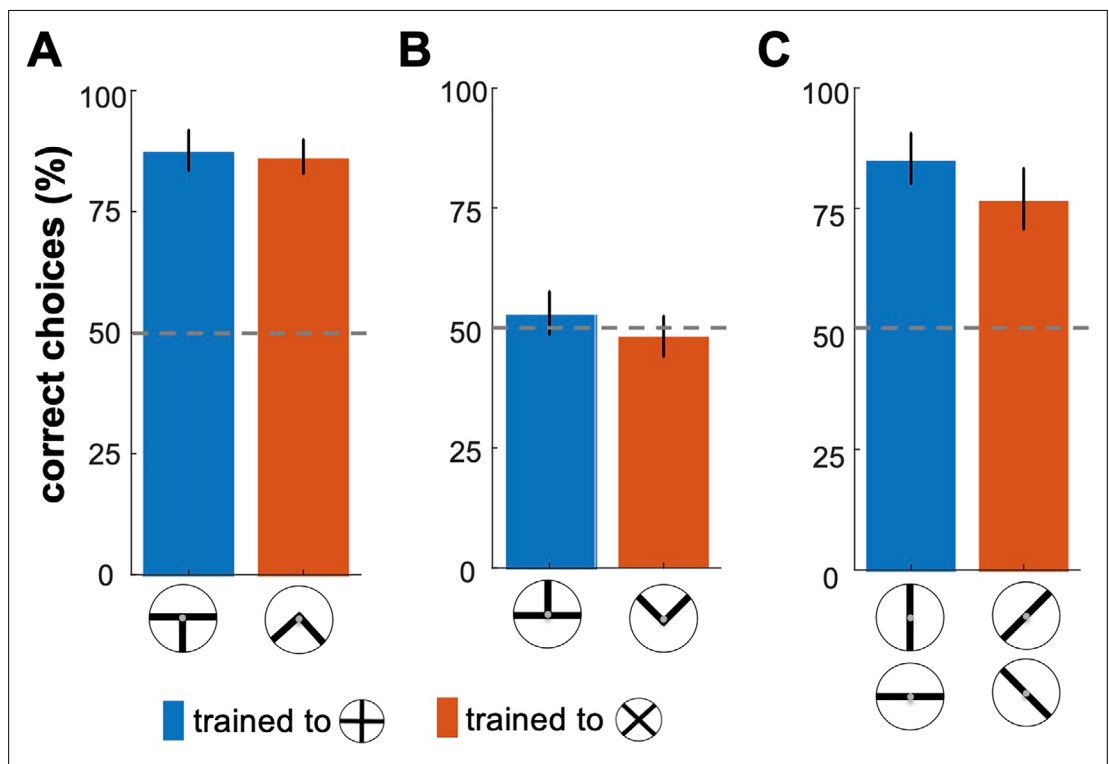

**Figure 2.** Bumblebees' performance in the transfer testing phase. This figure shows bumblebee performance in unrewarded tests where different sections of the training patterns were presented. (**A**) Bees successfully discriminated the bottom halves of the patterns, with statistically significant results (p < 7.7e-3). However, in (**B**), bees exposed only to the top halves did not show significant recognition of the correct patterns (p=0.83), suggesting these cues were insufficient for accurate discrimination. (**C**) shows that bees exhibited a significantly higher response to individual bars matching the rewarded orientations in the training (p<6.8e-4), indicating effective recognition based on specific visual cues learned during training.

The online version of this article includes the following figure supplement(s) for figure 2:

**Figure supplement 1.** Bumblebees' preference in the single-bar orientation test.

the patterns showed no significant preference for the correct pattern (*Figure 2B*; Wilcoxon signed rank test; z=0.17, n=10, p=0.83 for Group 1; z=0.17, n=10, p=0.85 for Group 2), indicating that these cues alone were insufficient for discrimination. Panel C displays the bees' responses to single bars from the training patterns. The single-bar orientation test showed a significantly higher response to bars matching the rewarded orientation across both groups (*Figure 2C*; Wilcoxon signed rank test; z=2.70, n=10, p=6.8e-3 for Group 1; z=2.6, n=10, p=9.3e-3 for Group 2). Furthermore, bees trained to the plus pattern favoured the vertical bar, while those trained to the multiplication pattern showed an equal preference for the 45° and –45° bars, indicating orientation-specific associations (*Figure 2— figure supplement 1*; Kruskal-Wallis test; Chi-sq=25.72; df = 39, p=1.0e-5 for Group 1; Chi-sq=19.0; df = 39, p=3.0e-4 for Group 1).

These results suggest that bumblebees rely on specific parts of the visual patterns—particularly the lower sections—and individual bar orientations to recognise the overall pattern. This finding implies that bees may focus on a discriminable local feature of a pattern that suffices for recognition, while disregarding other elements. This highlights the role of partial and component-based cues in bumble-bees' visual processing and learning.

## Bumblebee flight speeds and trajectories during the learning tests

To explore how bees choose the correct patterns and reject the incorrect ones, we analysed the bees' inspection behaviours, employing a custom algorithm to track bee locations and body orientations within each frame of the videos (*Figure 3A and B* and *Video 1*; see Video Analysis in Methods section).

The bees' initial flight speed upon entering the flight arena and approaching the first inspected stimulus was on median 0.20 (±0.13 SEM) m/s (*Figure 3C*). The speed reduced to a median of 0.11 (±0.10 SEM) m/s whilst inspecting the stimulus; the highest proportion of flight speeds was less than 0.1 m/s (*Figure 3D*). The bees' speed increased to a median of 0.20 (±0.24 SEM) m/s whilst traversing between the presented patterns (i.e. flights between the ⌀12cm x 10cm cylinder regions surrounding stimuli). Bees typically scanned the patterns (inspections with the lower flight speed of 0.20 (±0.13SEM) m/s) from a distance of 10 mm to 50mm from the stimuli (*Figure 3E*). The bees spent approximately 1.51 (±0.50 SEM) seconds inspecting a stimulus, irrespective of whether this was a plus sign or multiplication sign, or the correct or incorrect pattern (*Figure 4H*). The flight speed during inspection when rejecting a pattern was on average three times that of when the bee accepted a pattern and flew to the feeder (*Figure 4G*). However, analysis of the flight trajectories (*Figure 4A, C and E*) shows this was due to the bee accelerating away from the current pattern to the next one. Interestingly, the bees showed an overall tendency to scan the patterns with their bodies oriented at ~±30° relative to the rear stimuli wall, keeping one or other eye predominantly aligned to the stimuli during the scans (*Figure 4F*). Conversely, when flying between the patterns, they mostly looked forward in the direction of their motion with a much wider range of flight directions relative to the rear wall (see Discussion section).

## Bumblebees scanned specific regions of the patterns prior to making a decision

As the bees did not appear to be making pattern selections from a distance (*Figure 1E*, *Figure 1— figure supplement 1* and *Figure 3E*), we further analysed the movements of the bees whilst directly in front of the patterns. In most instances (Group 1 trained to ⊕: 89.2%, Group 2 trained to ⊗: 87%), the bees first traversed to, and then scanned, the lower part of the patterns regardless of whether the target was rewarding or aversive (*Figure 3A*). Each scan led to either a landing on the feeding tube (an acceptance) or the bee flying to another stimulus without landing (a rejection). The selection proportions of bees for each region of the patterns before making acceptance or rejection decisions are illustrated in *Figure 3F* for each group of trained bees. The bottom centre of the patterns attracted the highest proportion of interest among the five analysed regions. In Group 1, the bees scanned the bottom centre before deciding to land in 54% of all correct choices. In Group 2, the bees accurately rejected the patterns, and the bottom centre was scanned prior to their rejection in 39.7% of instances. Importantly, this proportion was comparable to the combined instances of scanning the lower left corner, the lower right corner, or both lower corners (totalling 47.5% for correct choices in Group 2 and 35% for correct rejections in Group 1). It is crucial to emphasise that the bees consistently

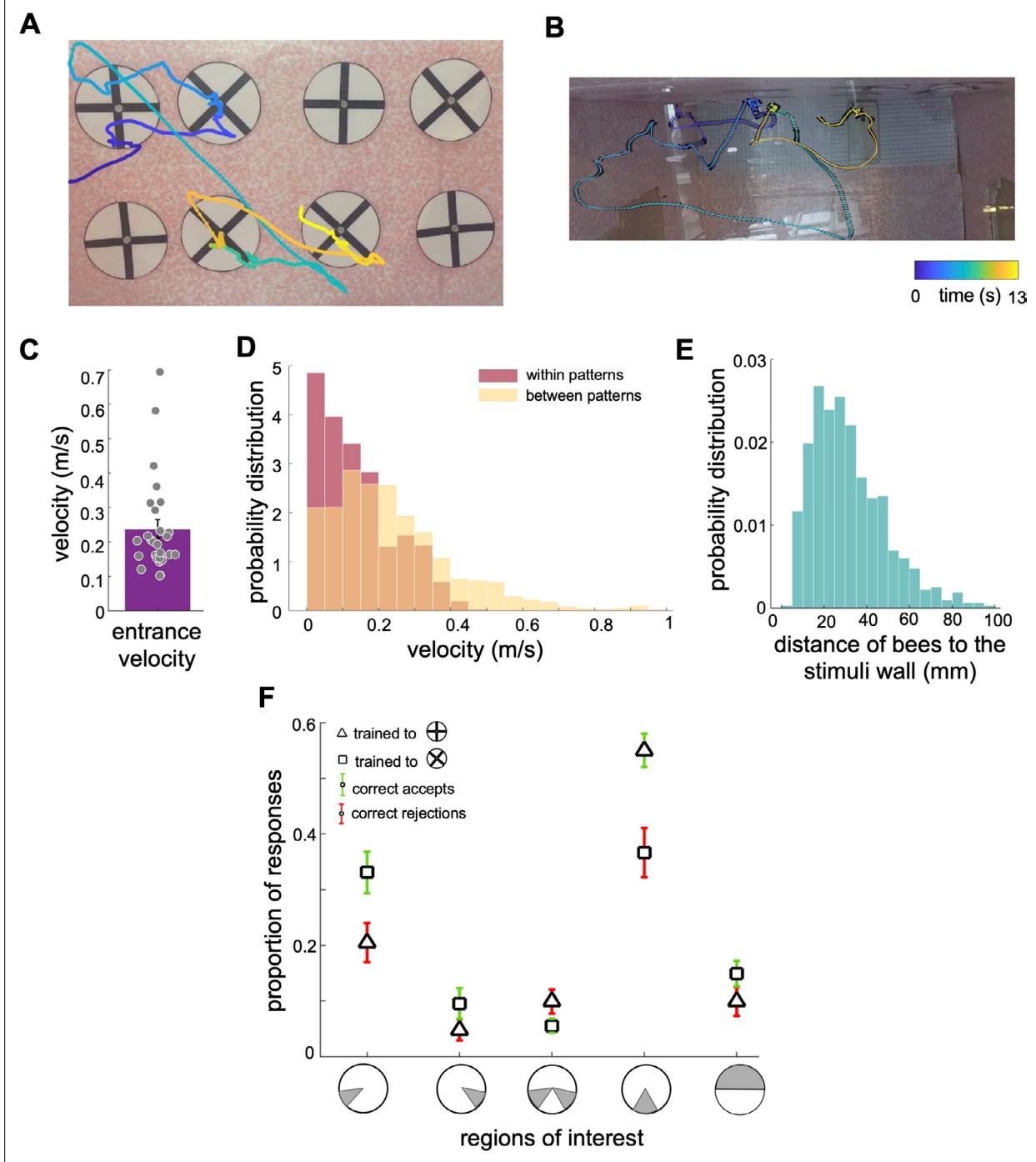

**Figure 3.** Scanning behaviour of bumblebees in a simple pattern discrimination task (A, B). An example of a flight path showing the activity of a bee during part of the learning test; presented bee trained to select the 'plus' and avoid the 'multiplication' patterns. Each point on the flight path corresponds to a single video frame with an interval of 4ms between frames which was recorded from the front camera (**A**) and top camera (**B**). The bee sequentially inspected each pattern, correctly landed on the multiplication sign and avoided the plus sign. The colour map changes from blue to yellow with increasing time (See *Video 1*). In (A) the plus patterns may appear slightly misaligned vertically due to visual distortion caused by the camera perspective and edge warping. In the actual experimental setup, the patterns were presented to the bees with a precise vertical alignment. In (B) the black lines in the right panel exhibit bee's body yaw orientations. (**C**) Distribution of average entrance flight speed toward the wall in the learning tests (See *Figure 1*). Filled dots: speed of each individual bee. (**D**) Probability distribution of the bees' speed in two conditions; when they were inspecting patterns (Red), and when they were flying between patterns (Yellow). This indicates that they slow down to scan patterns before accepting while they accelerate when they fly to another pattern. (**E**) Probability distribution of the bees' distance from the stimuli wall whilst inspecting patterns. (**F**) From video analysis, the proportion of scanned regions (mean ± SEM) of bees' inspections before the correct accept or correct rejection (x-axis: regions of

*Figure 3 continued on next page*

*Figure 3 continued*

interest are highlighted in grey). Triangles: inspection proportion (mean ± SEM) of Group 1 bees (trained to plus sign rewarding), squares: Group 2 bees (trained to multiplication sign), green error bars: correct accepts; red error bars: correct rejections.

displayed a preference for the lower left corner, which will be further discussed in the subsequent analysis. These preferences can be clearly seen on the heat map representation of the accumulated bee positions during scanning (*Figure 4B and D*). Bees trained on the protocol with the plus pattern as rewarding (Group 1) would typically approach the lower half of the stimulus (89% of inspections). They would scan the lower centre of the pattern (containing the vertical bar) and then fly directly to the pattern centre to access the feeding tube (*Figure 4A and E*, see *Video 2*). However, if the bees observed a multiplication sign, they would usually scan the lower left corner of the pattern, containing the +45°-angled bar of the multiplication sign (*Figure 4A*). Of these trials, over half consisted of a single corner scan before the bees rejected the patterns. A scan of the whole pattern was clearly not required: the inspection of a single diagonal pattern element was sufficient to ascertain that the pattern was not a plus sign. In the remaining cases the bees would traverse to the opposite lower corner, then scan the remaining oriented bar before rejection (*Figures 3F and 4A*). On average, in only 4.5% of such inspections did the bees only scan the right corner. Bees trained on the multiplication pattern (Group 2) showed a slightly different behaviour. If the bees were inspecting a multiplication sign stimulus, they would first approach the left or right lower section of the pattern (*Figure 4C and E*, see *Videos 3 and 4*). We still observed the same preference for the left-side inspections, double that of the lower right-side scans. However, there were far fewer instances of bees inspecting both corners before flying to the feeding tube (*Figure 3F*). When Group 2 bees (trained to the multiplication pattern) encountered a plus pattern they would again scan the lower centre at the base of the vertical bar (see *Video 3*). In contrast to the Group 1 bees accepting the multiplication symbol, these bees would also, on occasion, scan the lower left corner where no oriented bar was present (*Figure 4C*). However, the observed scanning behaviour is consistent with bees' performance in the transfer tests with having higher performance of availability of cue of the positive patterns in the lower part of the stimuli.

## Selective scanning and pattern recognition

*Figure 5* provides a detailed illustration of bumblebee scanning behaviour during the top-half and bottom-half tests, showing that bees focused their attention on specific pattern areas that they had learned to identify as minimal discriminative visual cues during training. Panels A and B display heat-maps of flight paths, with warmer colours indicating regions of heightened attention and prolonged hovering. In the bottom-half test (*Figure 5A*), bees trained to the plus pattern predominantly focused on the lower section of the vertical bar, while those trained to the multiplication sign concentrated on the space between the two diagonal bars in the bottom-half stimuli. These results indicate that bees rely heavily on the lower regions of the patterns as their primary source of visual information, and their scanning strategy remains consistent with the behaviour developed during training, even in the absence of the top sections of the stimuli (see comparison with *Figures 3F, 4B and D*). Conversely, in the top-half test (*Figure 5B*), despite the bees' inability to statistically discriminate between the presented patterns (as shown in *Figure 2B*), they continued to prioritise scanning the lower sections of the stimuli, even though these areas lacked any visual cues. Some bees expanded their search to the top half of the patterns and occasionally made incorrect acceptances or rejections after encountering vertical or diagonal visual cues. This behaviour highlights a strong bias toward learned lower regions for pattern discrimination and underscores the

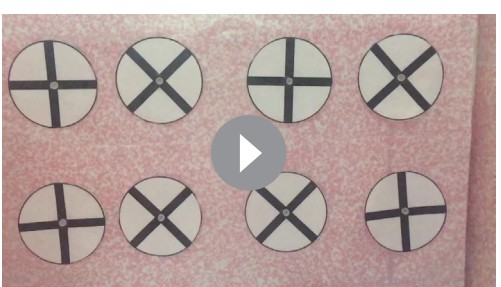

**Video 1.** Example video of a bee's scanning behaviour in the learning test. The bee was trained to find the reward from the multiplication. The bee inspected lower region of the patterns and rejected the plus and accepted several multiplication signs. The video was recorded by the front camera at 240 fps (frames per second).

https://elifesciences.org/articles/106332/figures#video1

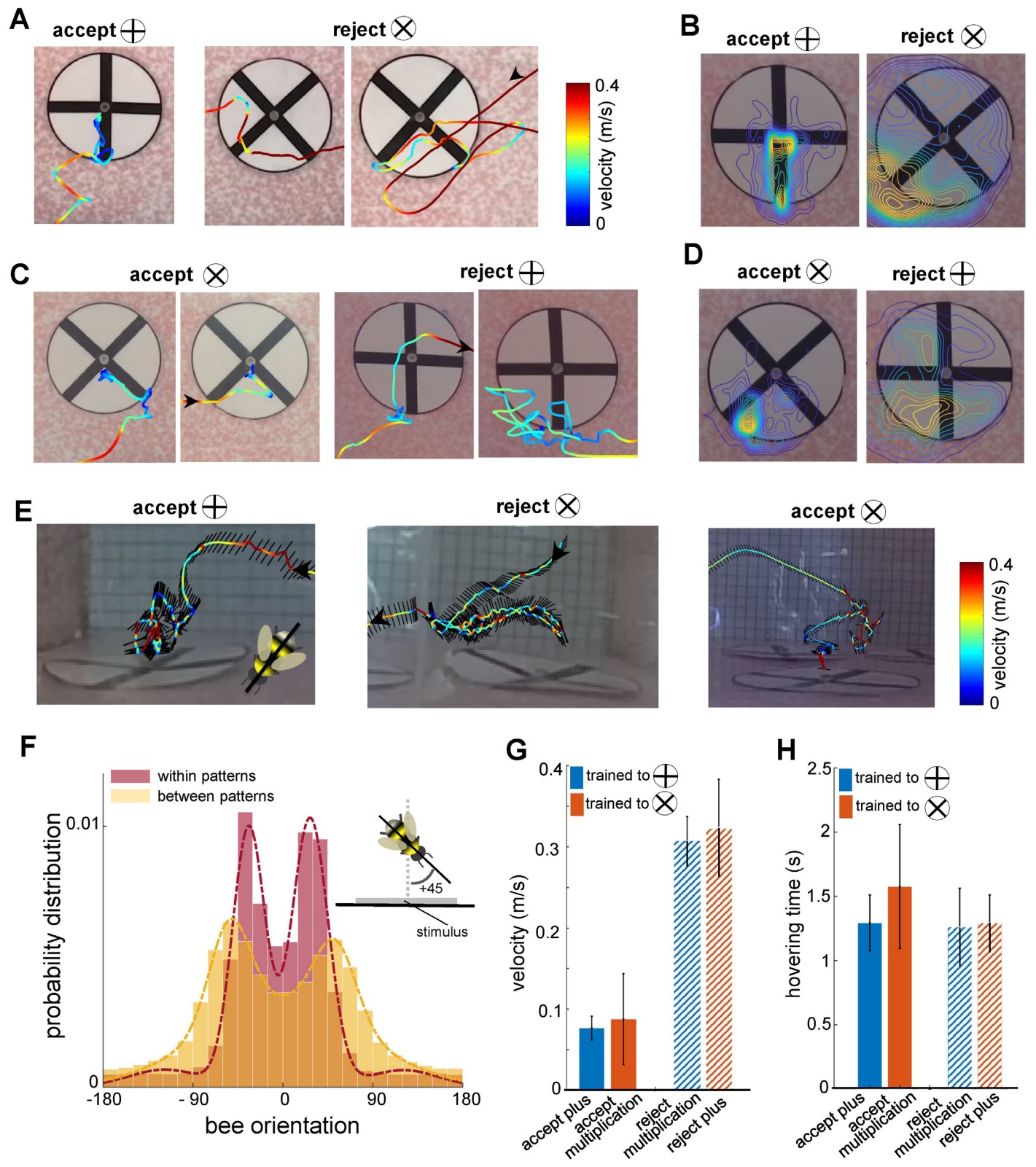

**Figure 4.** Bee scanning strategy in a pattern recognition task. (**A**) The flight paths of one example of acceptance and two examples of rejection behaviours of a bee trained to plus; the bee accepted the plus pattern after scanning the lower half of the vertical bar, while she rejected the multiplication pattern after scanning one or both diagonal bars. Line colour: flight speed 0.0–0.4 ms⁻¹ (see *Video 2*). (**B**) Group 1 (trained to plus) probability maps (heat-maps) of bees' locations per frame in front of plus and multiplication type stimuli during all learning tests. The yellow colours

*Figure 4 continued on next page*

*Figure 4 continued*

show the most visited regions. (**C** & **D**) same analysis as A, B for Group 2 bees trained to discriminate the multiplication sign from the plus sign. This indicates that bees typically scanned the lower half of the pattern with a lower speed to Group 1 bees, prior to their decisions (see *Videos 3 and 4*). (**E**) Three examples of bees' flight paths shown from the top camera; black lines show bees' body orientation during the flight, and arrows designate the start and ending time of scanning. (**F**) Probability distribution of the bees' body yaw orientation perpendicular to the rear stimuli wall in two conditions: when they were inspecting patterns (red) and when they were flying between patterns (yellow). Inset figure exhibits one example of a bee's orientation (viewed from the top) with +45° to the orientation of the wall on which stimuli were displayed. Bees viewed the patterns at a median ~±30° whilst scanning, with one or other eye having a predominant view. On the other hand, when they transitioned between patterns, the body orientation was more parallel to the flight direction with a wider distribution of orientations relative to the stimuli wall, resulting in a median of ~±50° to the stimuli wall. The dashed lines show the Gaussian mixture distribution models were fitted to each distribution (flights within patterns: $\mu_1 = +27, \mu_2 = -33$; flights between patterns: $\mu_1 = +51, \mu_2 = -55$). (**G**) Mean flight speed (±SEM) of scanning flight prior to decisions (accept and rejection) for both groups of bees. Blue: Group 1 (trained to plus sign); orange: Group 2 (trained to multiplication sign). (**H**) Inspection time (i.e., the time spent hovering in front of a pattern) for each symbol type for both groups of bees; inspection times of bees in front of both pattern types were equal regardless of their decision or training protocol.

influence of pre-learned scanning strategies and task-relevant visual features in guiding bumblebee decision-making, even under conditions where critical cues are absent.

As shown in *Figure 5C*, bees spent more time inspecting the lower parts of the patterns when accepting the visual cues associated with the correct patterns (Wilcoxon rank sum test; z=4.06, n=10, p=4.7e-5 for Group 1; z=3.64, n=10, p=2.7e-4 for Group 2). In contrast, when rejecting incorrect patterns, bees exhibited shorter hovering times in the bottom sections of both the bottom-half and top-half stimuli (Wilcoxon rank sum test; z=9.66, n=10, p=4.1e-22 for Group 1; z=9.19, n=10, p=3.6e-20 for Group 2), suggesting they rejected the stimuli quicker when they could not locate positive visual cues in the lower section. However, there was no significant difference in hovering time when bees correctly rejected the negative patterns (Wilcoxon rank sum test; z=1.89, n=10, p=0.057 for Group 1; z=0.83, n=10, p=0.4 for Group 2). This behaviour aligns with the equal selection of both types of stimuli in the top-half test (*Figure 2B*), where bees were unable to find familiar visual cues in the lower sections, leading to random choices or rejections. These findings highlight a scanning strategy in which bees rely on the lower regions of patterns for cue-based decision-making, a behaviour developed during training in response to positive and negative patterns, regardless of the cue's actual location within the stimuli during inspection.

## Discussion

In this study, we explored the flight characteristics and active vision strategies used by bumblebees (*Bombus terrestris audax*) to solve a simple yet challenging visual discrimination task, in which bees were required to distinguish between a multiplication sign and its 45°-rotated variant (a plus sign). Previous studies showed that honeybees (*Apis mellifera*) were unable to discriminate these patterns when prevented from viewing them up close (*Srinivasan, 1994*; *Srinivasan et al., 1994*). However, our results demonstrate that bumblebees, within our flight arena, could successfully learn to distinguish between these patterns while they were allowed to freely inspect the patterns. Bumblebees in our study preferred to inspect both rewarding and aversive stimuli from a close distance (1–5 cm), with

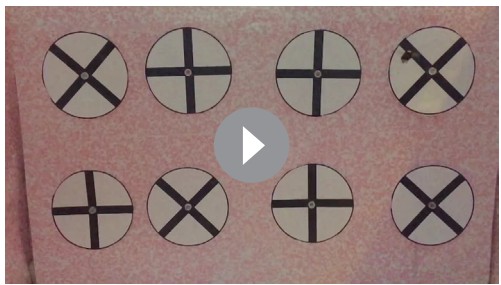

**Video 2.** Example video of rejecting the multiplication and accept the plus. The video was recorded at 240 fps (frames per second).

https://elifesciences.org/articles/106332/figures#video2

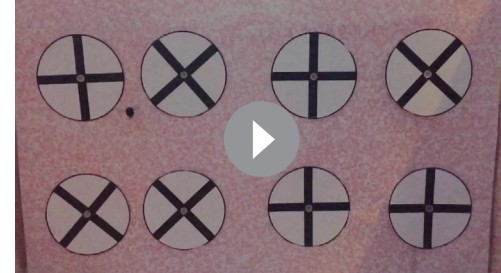

**Video 3.** Example video of rejecting two plus signs and accept the multiplication sign. The video was recorded at 240 fps (frames per second).

https://elifesciences.org/articles/106332/figures#video3

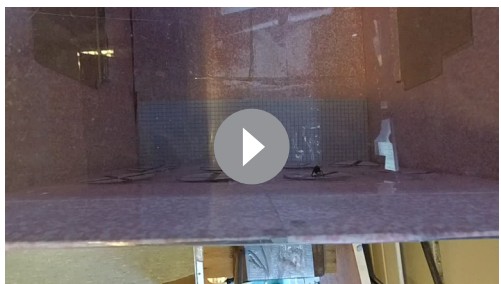

**Video 4.** Example video of from the top camera. The video is the same flight track as *Video 3* but recorded from the top camera. The video was recorded at 120 fps (frames per second).

https://elifesciences.org/articles/106332/figures#video4

the patterns subtending a visual angle of approximately 90° to 160°, in contrast to the ~50° angle used in honeybee experiments (*Srinivasan et al., 1994*). Interestingly, upon entering the arena, bees did not show a preference for initial stimulus selection from a distance, with approximately 50% of initial inspections directed toward incorrect patterns (*Figure 1E*), a behaviour consistent with previous reports on bee visual concept learning (*Guiraud et al., 2018*). Even when they chose correctly, bees consistently performed a close-range scan of pattern elements before landing on the feeder (*Figures 3 and 4*). These findings suggest that bumblebees rely on close inspection of specific features within a pattern to make accurate acceptance or rejection decisions. For this experimental paradigm, it appears that bees do not make stimulus selection decisions from a distance but instead choose to approach and inspect patterns closely before committing to a choice. This behaviour supports the hypothesis that bumblebees employ a sequential feature-scanning strategy, enabled by active vision, to efficiently distinguish between complex visual stimuli in a controlled environment.

Our experimental paradigm cannot confirm with certainty that bumblebees are unable to discriminate these simple patterns from a distance; for that we would need to control for distance as done with the honeybee experiments (*Horridge, 1996*; *Srinivasan et al., 1994*). However, our experiment allowed us to carefully analyse the bees' scanning behaviour of visual features and to extract useful insights into the active vision of bees.

In brief, our bumblebees had no difficulty in learning to identify and associate either the plus or multiplication signs with reward, with all bees achieving over 90% accuracy after 70 trials (*Figure 1C*). This performance was preserved during the unrewarded learning tests (*Figure 1D*). Our bespoke video analysis toolkit allowed us to track the bee positions and body yaw orientations for every frame of each learning test. The most notable, and consistent, characteristics observed were:

## Partial pattern inspection

The bees primarily flew to, and scanned, the lower half of the patterns (*Figure 2F*). This suggests that the lower half was all the bees learned. Indeed, when exposed to a transfer test with only the top half of the pattern available, bees failed to identify the correct halves of the training patterns (*Figure 2E*). A previous study showed that honeybees (*Apis mellifera*) trained in a Y-Maze using absolute conditioning (where only the positive pattern and a secondary blank stimulus is provided) assigned more importance to the lower half of the pattern to that of the top half (*Chittka et al., 1988*; *Giurfa et al., 1999*). During tests with only the top half of the training pattern and a novel pattern they failed to select the correct pattern half. Conversely, if bees were presented with the lower half of the training pattern and again a novel pattern, they could identify the correct stimulus. In contrast, when trained using differential conditioning (using both rewarded and unrewarded patterns), the honeybees learned the whole pattern; correctly identifying both bottom and top half patterns during tests. However, in this instance, unlike in our study, the bees' choices were recorded from a distance (for apparatus details, see *Horridge, 1996*) and bees' flights were not analysed systematically. However, in our study, we did not test bumblebees with just the lower half of the pattern. It is possible that just presenting the lower section of the pattern would also decrease performance. Future work is required to address this limitation. In addition, it would be useful to see the bee performance if only the left or right side of the pattern is presented (see below).

In a more recent study, in which the flight path of bees was also analysed, *Guiraud et al., 2018* showed how honeybees (*Apis mellifera*) can solve a conceptual learning task of 'above and below' by scanning the lower of two pattern elements presented on the stimuli; this provided sufficient information for the bees to make a decision without needing to understand, or inspect, the relationship between the top and bottom pattern elements.

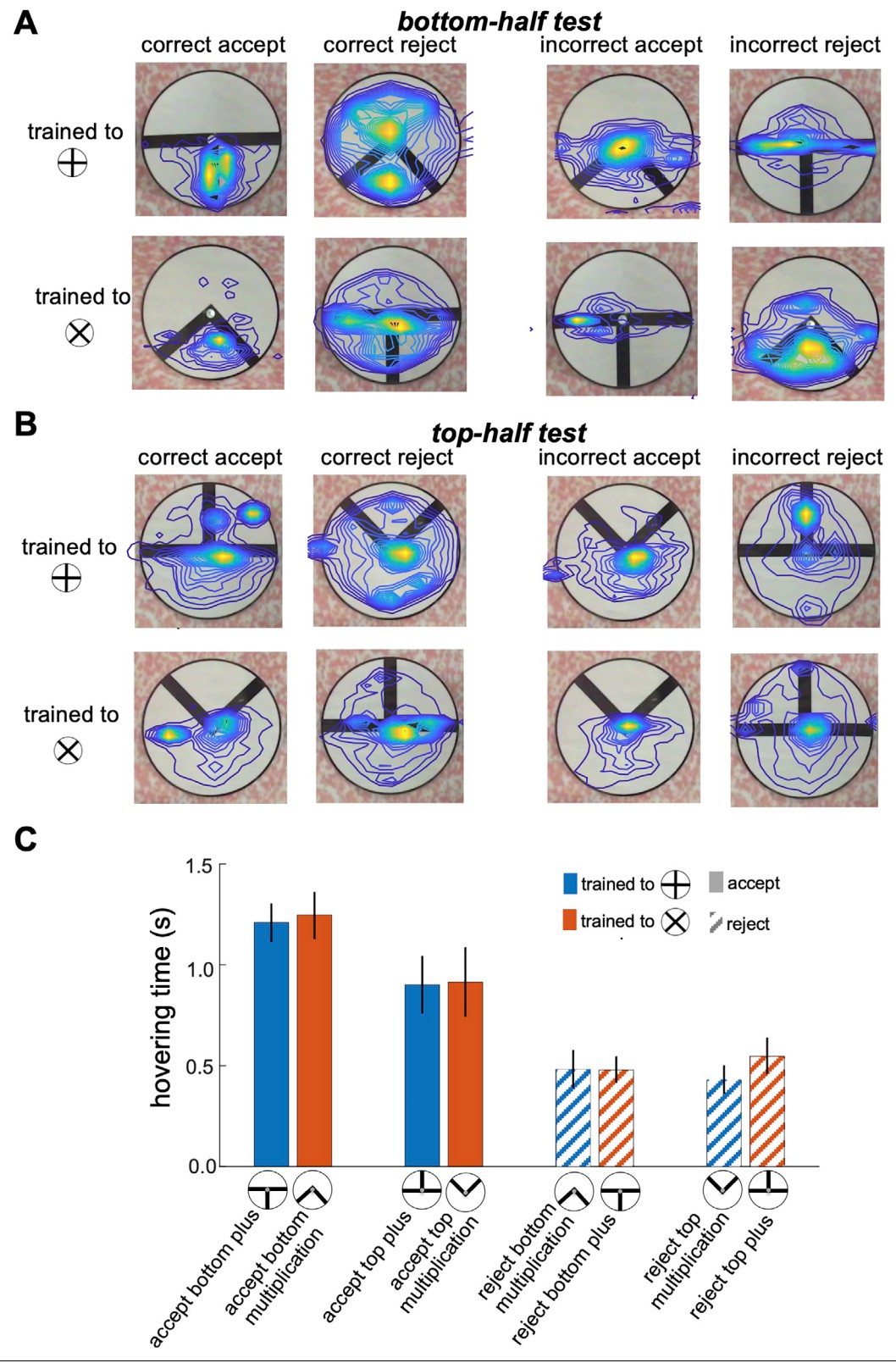

**Figure 5.** Bumblebee scanning behaviour when inspecting partial and component visual cues. (**A & B**) Heatmaps of bumblebee flight paths overlaid on the visual stimuli during the bottom-half test (**A**) and top-half test (**B**). Each panel is grouped based on the bees' decisions to correctly or incorrectly accept or reject the presented test patterns. The heatmaps illustrate areas where bees spent the most time hovering, with warmer colours indicating

*Figure 5 continued on next page*

*Figure 5 continued*

prolonged hovering and increased attention to specific regions of the pattern. The contour lines further highlight the spatial distribution of bee activity over the stimuli. (**C**) presents bar plots comparing hovering times for correct accept and correct reject responses. The results reveal a strong focus on the bottom sections with longer hovering time when bees correctly accepted the positive pattern (p<2.8e-4), suggesting a scanning strategy that prioritises these areas for pattern discrimination. However, there was no significant difference in hovering time when bees correctly rejected the negative patterns (p>0.05). This preferential focus on specific regions implies that bees adopt scanning behaviours centred on discriminative visual cues, particularly in the lower half of the patterns, as an effective strategy for pattern recognition.

---

The reason why bees primarily focus on the lower half of the pattern instead of the top half remains unclear. However, several factors may contribute to this behavioural preference. Flowers positioned at the lower end of an inflorescence are typically older, larger, and contain more nectar compared to those located higher up (*Davies et al., 2012*; *Heinrich, 1979*; *Heinrich, 1979*; *Heinrich, 1975*; *Johnson and Anderson, 2010*). As a result, bees can optimise their nectar collection efficiency by starting from the bottom and gradually moving upward. Additionally, studies have indicated functional differences in bee eye regions (*Lehrer, 1998*; *Spaethe and Chittka, 2003*; *Wakakuwa et al., 2005*), suggesting specialised neural mechanisms in different eye regions that serve the specific needs of bees. If the ventral eye region provides a neural advantage, bees can still scan the top half of patterns by flying above the stimuli and examining them with their ventral eye regions. Another possibility is that the lower part of the pattern may contain distinguishing visual features or fractures that provide sufficient information for bees to solve the visual task. By scanning the lower part of patterns, bees can capture the key information necessary for successful pattern recognition, thereby enhancing their speed in solving the task (*Chittka et al., 2009*; *Guiraud et al., 2025a*; *MaBouDi et al., 2023*). Additionally, inspecting from the lower part of the pattern and moving upward through flight may be easier for the motor system of bees.

## Initial side preference

The bees had a significant preference for initially scanning the left side of the multiplication pattern (*Figure 2F*). This left side preference for visual objects, known as pseudoneglect, is also seen in humans (*Jewell and McCourt, 2000*), and birds (*Diekamp et al., 2005*; *Rugani et al., 2015*). This preference may allow an individual to always start its inspection of a stimulus at the same location, allowing for consistent learning and recognition of natural stimuli; but it remains a curiosity as to why the left preference was so prevalent amongst the bees tested (*Anfora et al., 2011*; *Giurfa et al., 2022*; *Letzkus et al., 2008*; *Figure 2F*). In humans and birds, this lateralisation of spatial attention may have evolved once in a common ancestor (*Diekamp et al., 2005*). However, since the visual system of insects evolved largely independently from that of vertebrates, the left-side bias must have emerged by convergent evolution. Its computational neural advantage in bees or vertebrates (if any), is not known.

## Common body orientation during scans

The yaw orientation of the bees' bodies was most often at ~±30° to the stimuli during pattern inspections. In this manner, one or other of the bees' eyes would face the pattern, with only a small proportion of the opposite eye having visual access to the pattern. There was no overall preference for the left or right eye (with median orientations at ~−33° and ~+27°, respectively) during scans (*Figure 4F*). In previous modelling work, *Roper et al., 2017* showed that lateral connections from both the left and right lobula to the bee mushroom bodies allowed for better pattern recognition during partial occlusion of stimuli. However, this came at the expense of fine detail recognition. Therefore, and counterintuitively, having one eye mostly obscured from the pattern may provide the mushroom bodies (learning centres of the bee brain) with more distinct neural inputs. It may also allow bees to learn both the pattern and location cues simultaneously whilst scanning a resource. Future work will be needed to see if this behaviour is particular to the patterns used in this experiment, or a stereotypical behaviour.

## Commonality in scan strategies is based on stimuli, not protocol

It may seem sensible, from the bees' perspective, if trained on plus signs, only to inspect the lower centre of the pattern for the vertical bar. However, both groups of trained bees initially approached and scanned the plus and multiplication signs in the same manner, typically checking the lower left corner of the multiplication sign and the vertical bar of the plus sign. This might suggest that the bees did not learn the relative position of the cues and simply searched for the first visual item at the lower left of the pattern. However, the flight tracking analysis conflicts with this hypothesis. Bees initial scan of the left 45° bar of the multiplication symbol was occasionally followed by a scan of the adjacent 45° bar on the right of the multiplication pattern. Additionally, with the group trained on multiplication as rewarding, after scanning the vertical bar of the plus they occasionally flew to the lower left corner; presumably checking for the multiplication sign-oriented bar. We therefore assume that the stimulus directs the scanning behaviour of the bee, and in turn the bee is learning both rewarding and aversive pattern features during training. Previous works have shown that bees will typically follow the contours of a stimulus during inspections, but can also learn to suppress this strategy if needed to solve a given task (*Lehrer, 1994*; *Lehrer and Srinivasan, 1994*). Further experiments will be required to ascertain the particular rules which dictate the bees' flight manoeuvres based on the stimuli provided.

In the pioneering works of Karl von Frisch, 1914, free-flying bees were trained to find sugar reward on certain black or coloured patterns placed horizontally on a white background. Later studies showed that bees only used local cues corresponding to their approach direction when the stimuli were presented to them horizontally (*Wehner, 1967*). Since bees were not able to capture global shapes in this position, this might be the reason bees could only recognise some simple patterns in the early studies. However, vertical presentation of stimuli was developed to examine what diversity of visual features bees may use, such as orientation (*van Hateren et al., 1990*), radial, bilateral symmetry (*Giurfa et al., 1996*; *Horridge, 1996*), or spatial frequency and ring-like structures (*Horridge and Zhang, 1995*). To control the decision distance and understand which cues were utilised by bees to recognise the target pattern, Y-mazes were used (*Srinivasan and Lehrer, 1988*). Although such mazes enabled researchers to control the cues that bees could see when making decisions about visual patterns from a distance, it is a less useful paradigm to inspect the scanning strategies used by bees.

Analysing rejection responses provides valuable insights into the decision-making processes and factors that influence precise visual assessments. This type of analysis not only reveals the criteria and thresholds that contribute to decision-making strategies but also helps uncover the main visual features that bees rely on to enhance their performance (*Chittka et al., 2009*; *Green and Swets, 1966*; *MaBouDi et al., 2023*). Surprisingly, rejection responses have been largely ignored in the existing literature of insect visual learning. In this study, we aimed to bridge this gap by analysing both acceptance and rejection behaviours in bees. Our results uncovered distinctive responses exhibited by bees when accepting or rejecting a pattern, indicating that both positive and negative patterns play a role in shaping their scanning behaviour. These findings shed light on the importance of negative patterns in understanding visual discrimination and provide substantial contributions into the comprehensive nature of bees' visual processing capabilities.

In our study, we observed that bees employed shorter scanning times in rejecting patterns when presented with only the bottom-half or top-half of the stimuli, compared to their scanning behaviour during pattern acceptance (*Figure 5C*). However, the hovering times for acceptance and rejection behaviours were approximately the same when the observed pattern matched their expectations from the learning tests (*Figure 4F*). This behaviour aligns with findings from *MaBouDi et al., 2023*, who reported that bees make rapid rejections when learned evidence is reduced. These observations suggest that bees dynamically utilise both sampling and decision-making strategies to optimise foraging efficiency, actively integrating sensory information with higher-level processes, such as decision-making, in visual discrimination tasks.

The results of this study demonstrate that bumblebees exhibit a persistent, refined scanning strategy developed through training, focusing primarily on the lower sections of patterns for effective cue-based decision-making (*Figures 2F and 3*). This preference for specific cues persists even when bees are presented with altered patterns (*Figure 5A*), as bees trained to associate distinct visual cues with positive or negative outcomes consistently prioritised the lower regions of both familiar and modified patterns during inspection. For example, in the bottom-half test, bees successfully generalised their learned associations to cues in the lower sections, achieving statistically significant discrimination

(*Figure 5A*). Interestingly, even in the top-half test, bees continued to focus on the lower sections despite the lack of any visual cues in those areas (*Figure 5B*). This inability to generalise their learned scanning strategy to the top-half test reveals critical limitations in their decision-making process, as bees failed to adapt their scanning behaviour to identify discriminative features in the top halves of the patterns. This suggests a strong bias towards pre-learned spatial regions, reflecting a reliance on fixed scanning strategies that prioritise previously reinforced visual cues in the lower sections, even when these cues are absent. By reducing their focus to minimal, stable visual cues, bees appear to adopt a strategy that minimises cognitive load, simplifying decision-making in complex visual environments (*Guiraud et al., 2025a*). These findings indicate that bumblebees' pattern recognition relies on a spatially selective scanning approach that becomes robust against changes in pattern structure, while also emphasising the importance of localised, learned visual features in guiding their decision-making processes. However, the observed limitations highlight that their scanning strategy is highly experience-dependent, restricting adaptability when visual information is presented outside familiar regions. Nevertheless, this result supports the hypothesis that bees actively seek out stable, minimal cues to streamline recognition and maximise efficiency in complex visual environments.

In recent years the availability of high-speed camera equipment and computer processing power has enabled closer examination of the flight trajectories of insects (*Doussot et al., 2020*; *Lent et al., 2013*; *MaBouDi et al., 2023*; *Mamiya et al., 2011*; *Mamiya et al., 2011*; *Odenthal et al., 2020*; *Philippides et al., 2013*; *Stowers et al., 2017*). In this study, we showcase a suite of tools for automatic video tracking of bees in free flight and during their scanning manoeuvres, as well the algorithms needed to analyse and visualise the large amount of positional and orientation data these tracking produces. In our previous work on 'above and below' conceptual learning (*Guiraud et al., 2018*) we had to manually view and annotate 368 hr of video footage (46 hr of video footage taken at 120 fps, watched at 1/8th speed). In contrast, here the only manual process was providing a mask frame (without the bee present) per test and marking the feeder positions within that frame. Notably, the introduced algorithm can automatically track bees and analyse their flight without requiring prior training on the data and operates with almost no free parameters. This results in a faster, energy-efficient algorithm for tracking animal behaviour using affordable, standard, or high-frame-rate cameras. With only a small number of test videos to process this was not an issue, but even here, recent advances in making convoluted neural networks for pattern recognition accessible to non-programmers (playground. tensorflow.org, https://runwayml.com/), as well as the more research programmer-centric DeepLabCut (*Nath et al., 2019*), allows researchers to provide a few dozen labelled mask frames and have these systems process thousands of mask images for all the other videos (*Egnor and Branson, 2016*). Future work is proposed that would record the flight paths of the bees throughout the whole training paradigm; this would allow us to determine, if, and how, the bees' strategy and flight trajectories evolve over time. This would produce substantially more flight recording data than analysed here, but our suite of tools presented here will vastly reduce the workload. Similarly, the ability to visualise either individual flight paths (*Figure 4A*) or combined heat maps of positional data (*Figure 4B*) allowed us to quickly identify behavioural aspects of interest. Histograms of velocity, distance and orientation can be quickly generated, but more importantly the parameters defining the areas of interest can be modified and processed in a matter of minutes. Previous studies have relied upon binary fixed decision lines (*Avarguès-Weber et al., 2012*; *Horridge, 1996*; *Horridge and Zhang, 1995*; *Lehrer and Srinivasan, 1994*), with experimenters manually recording these limited behavioural data. Our in-depth analysis on such a straightforward pattern recognition task highlighted key behavioural characteristics, which can now influence future work on active vision, this simply would not have been viable without these automated tools.

## Materials and methods
### Animals and experimental setup
In total, forty bees from six colonies of bumblebees (*Bombus terrestris audax*, purchased from Agralan Ltd., Swindon, UK) were used during this study. Colonies were housed in wooden nest boxes (28 x 16 x 11 cm) connected to a wooden flight arena (60 x 60 x 40 cm) via an acrylic tunnel (25 x 3.5 x 3.5 cm). The arena was covered with a UV-transparent Plexiglas ceiling (*Figure 1A*). Illumination was provided from directly the above the arena via high frequency fluorescent lighting (TMS 24 F lamps with HF-B

236 TLD ballasts, Phillips, Netherland; fitted with Activa daylight fluorescent tubes, Osram, Germany) enclosed within a light diffuser box; the flicker frequency of the lights was ~42 kHz, which is well above the flicker fusion frequency for bees (*Skorupski and Chittka, 2010*; *Srinivasan and Lehrer, 1984*). The walls of the arena were covered with a Gaussian white and pink pattern (MATLAB generated); this provided good contrast between the colour of the bees and the background, required for the video analysis. Sugar water was provided at night through a mass gravity feeder and removed during the day when bees were performing experiments to ensure motivation. Pollen was provided every 2 days into the colonies.

## Stimuli

The stimulus patterns were printed on laminated white discs (10 cm in diameter) to allow for easy cleaning with a 70% ethanol solution between training bouts and tests. The training patterns consisted of two black bars (1x10 cm) presented in two configurations: (1) *Plus pattern:* one vertical and one horizontal bar aligned at their centre (⊕); and (2) *Multiplication pattern:* the same configuration as the plus pattern but rotated by 45° to form an 'X' shape (⊗). Additional patterns were created for transfer tests, which presented only the top half (top-half test) or the bottom half (bottom-half test) of the training stimuli to assess partial shape recognition. Furthermore, single-bar patterns were included to test orientation discrimination, with bars oriented vertically, horizontally, at 45°, or at –45°. Each pattern included a 2 mm black margin around the outer edge of the disc to ensure consistent visual boundaries.

Each disc was attached at its centre to the back wall of the arena via a feeder made from a small 0.5 ml Eppendorf tube without a cap (5 mm in diameter). The feeder held 10 µl of one of three solutions: 50% sucrose solution (w/w) as a reward, saturated quinine solution (0.12%) as an aversive stimulus, or sterilised water as a neutral control. This setup allowed for controlled delivery of reinforcement during training and testing, enabling us to systematically investigate the bees' ability to distinguish between complete, partial, and orientation-specific stimuli and to analyse their visual scanning behaviour in making acceptance or rejection decisions.

## Training and test protocol

Prior to the experiments, bumblebees could freely fly between the colony and a gravity feeder providing 30% sucrose solution (w/w) placed in the centre of the flight arena. Successful foragers were individually marked on the thorax with number labels (Opalithplättchen, Warnholz & Bienenvoigt, Germany) for identification during the experiment. Every day of experiment, marked bees were randomly selected and pre-trained to receive 50% sucrose solution from eight white discs presented on the rear wall of the area. These pre-training stimuli were 10 cm in diameter with 2 mm wide black margins at the edges. After several bouts of pre-training, a forager that learned to take the sucrose from the feeder at the centre of the white pattern was selected for the individual experiment. During training, only the selected bee was allowed to enter the flight arena.

To improve the accuracy and the speed of learning, a differential conditioning protocol was used. Four multiplication and four plus pattern stimuli were randomly affixed to set positions on the rear wall of the arena. Each stimulus was 3–6 cm horizontally, and 5 cm vertically separated from the next stimulus, or arena wall/floor/ceiling (*Figure 1B*). One group of bees (n=10) was trained to receive 10 µl 50% sucrose solution (w/w) from the feeding tubes at the centre of the plus pattern stimuli, and to avoid the multiplication patterns that contained 10 µl saturated quinine solution. The second group (n=10) was trained on the reciprocal arrangement, that is associate the multiplication pattern with a reward and avoided the plus pattern.

Bees were allowed to freely choose and feed from multiple stimuli, until they were satiated and returned to their hive; empty tubes were refilled with 10 µl of sucrose solution after the bee had left the correct stimulus and made its next choice. A bout of training was completed once the bee returned to the hive. After each bout, all feeding tubes were cleaned with soap and 70% ethanol and then rinsed with water. The patterns were separately washed with 70% ethanol. Both tubes and patterns were air-dried in the lab before reuse. The position of stimuli on the wall were randomly varied for each bout to prevent bees from using the location of the reward when solving the task.

After five bouts of training, or upon reaching 80% performance in the last 10 choices, the bees were subjected to four tests to evaluate whether and how they could recognise and select the correct pattern (*Figure 1B*).

1. *Learning test:* Bees were presented with the same multiplication and plus patterns used during training. This test aimed to verify that the bees had learned to associate the correct pattern with the reward and to control for any possible olfactory cues that may have been used during training.
2. *Bottom-half transfer test:* Bees were exposed to stimuli presenting only the bottom half of the multiplication and plus patterns. This addition aimed to evaluate whether the bees could recognise the correct pattern using only the lower part of the shapes.
3. *Top-half transfer test:* Bees were exposed to novel stimuli that presented only the top half of the multiplication and plus patterns (see *Figure 1B*). This test was designed to determine if the bees could still recognise the correct pattern based solely on the top half.
4. *Single-bar orientation test:* The fourth test introduced single-bar stimuli in four different orientations: vertical, horizontal, 45°, and –45°. This test assessed whether the bees could distinguish the pattern based on a single bar orientation rather than the full or partial pattern configurations.

For the first three tests, as during training, both correct and incorrect stimuli were presented, with four instances of each pattern randomly positioned on the rear arena wall. In the single-bar orientation test, two patterns of each of the four orientations were presented to the bees. Each stimulus feeding tube was filled with 10 µl of sterilised water (i.e. no reward or punishment) to prevent reinforcement effects during testing. One to two refresher bouts of training (with reward and punishment), identical to the training bouts, were conducted between tests to maintain the bees' motivation. The sequence of the tests was randomised for each bee to minimise potential order effects. The first 20 bees were tested only with the first two tests. Additionally, some bees did not complete all tests due to a loss of motivation to return to the flight arena. The entire pre-training, training, and testing process was conducted continuously, taking approximately 3–5 hr per bee.

## Video analysis

The arena was equipped with two cameras to record all activity of bees during tests. An iPhone 6 (Apple, Cupertino, USA) with 1280x720 pixels and 240 fps (frames per second) was positioned above the arena entrance tunnel viewing the rear stimuli wall, filming the bee's flight in front of the stimuli wall and patterns. The second camera, a Yi sport camera (Xiaomi Inc China) with 1280x720 pixels at 120 fps, was placed on the top of the rear wall orientated downward to view the stimuli. The first 120 s of each test were recorded and analysed.

To analyse bees' scanning behaviours in front of the stimuli, prior to their choices, a MATLAB algorithm was developed that detected the bees automatically and then tracked the centroid of the bee bodies within each frame as they flew through the arena. For each frame, the algorithm subtracted a background mask image to find new candidate positions of the bee using MATLAB's blob detection function. The parameters of this function were set to detect the blob with the same approximate size of a bee. In addition, an elliptic filter was used in the frames from the top camera to extract the bees' body orientations. We utilised the MATLAB smoothing function ('filter') to exclude any erroneous data points and correct trajectories. Examples of the annotated flight paths and corresponding video recordings are shown in *Figure 2* and *Video 1*.

Using the first frame of each video recording, we manually specified the x, y pixel position of each of the eight pattern centres (i.e. entrances to the feeding tubes). The speed of the bees was calculated using the calibration data, the distance of the camera to the stimuli, and the pixel positions of the bee's trajectory, converted into real-world units based on the known dimensions of the experimental setup. After calculating the speed of each bee at each point of the trajectory, a threshold rule was applied to the trajectories close to the feeding tube positions to identify if the bees had landed, labelling the decision as either a correct/incorrect accept or rejection. This 'landing' threshold was determined by K-means clustering (*MaBouDi et al., 2023*; *MaBouDi et al., 2020a*) of all bee speeds within the specified region of the feeding tubes. For further analysis of stimuli inspections, the flight speeds, distances from wall, orientation, inspection times, areas of interest, and heat maps, we extracted the bees' trajectory data (using the above procedure) from a cylindrical region in front of each stimulus, with a diameter 12 cm around the pattern centre and 10 cm out from the stimuli wall. Bespoke MATLAB algorithms were developed to calculate and plot the required datasets for each of these individual stimuli analyses (see examples: *Figures 2 and 3*). Unfortunately, one of the learning test videos from Group 1 (trained to plus) was accidently recorded at just 30 fps; we therefore excluded

it from the above flight analysis. This video was sufficient, however, for the behavioural results of the choices and rejections of the bee to be extracted.

### Statistical analysis

A generalised linear mixed model (GLMM) with Binomial distribution and link function 'logit' was applied to the bees' choices recorded during the training phase to evaluate the effect of colony and group on bees' performance and compare the learning rate between two groups of bees. To assess the bees' performances in the tests, we analysed the proportion of correct choices for each individual bee. The proportion of correct choices was calculated by the number of correct choices divided by the bee's total choices during the first 120 s of the test. A choice was defined as when a bee touched a microcentrifuge feeding tube with her antennae or when she landed on a feeding tube. We then applied the Wilcoxon signed rank or Wilcoxon rank-sum tests to compare the bees' responses to the learning test and the transfer test.

## Computing environment

All data analysis and visualisation were performed using MATLAB (RRID:SCR_001622) and Python (RRID:SCR_008394). Both Python and MATLAB were used for image processing, video analysis, and custom script development. MATLAB was also used for statistical analysis and behavioural data analysis.

## Acknowledgements

We thank Olivia Brookes for her help in collecting the preliminary data. This study was supported by the Human Frontier Science Program (HFSP) grant (RGP0022/2014), the Engineering and Physical Sciences Research (EPSRC) Programme Grant Brains on Board (EP/P006094/1) and Horizon Europe Framework Programme grant NimbleAI - Ultra energy-efficient and secure neuromorphic sensing and processing at the endpoint. MGG was supported by ARC Discovery Projects (DP230100006 and DP210100740) and Templeton World Charity Foundation Project Grant (TWCF-2020–20539).

## Additional information

### Competing interests

Mark Roper: Employee of Ben Thorns Ltd. The other authors declare that no competing interests exist.

### Funding

| Funder | Grant reference number | Author |
| --- | --- | --- |
| Human Frontier Science Program | RGP0022/2014 | HaDi MaBouDi<br>Marie-Geneviève Guiraud<br>Mark Roper<br>Lars Chittka |
| Engineering and Physical Sciences Research Council | Programme Grant Brains on Board (EP/P006094/1) | HaDi MaBouDi<br>James AR Marshall<br>Lars Chittka |
| HORIZON EUROPE Framework Programme | | Lars Chittka |

The funders had no role in study design, data collection and interpretation, or the decision to submit the work for publication.

### Author contributions

HaDi MaBouDi, Conceptualization, Data curation, Software, Formal analysis, Supervision, Validation, Investigation, Visualization, Methodology, Writing – original draft, Project administration, Writing – review and editing; Jasmin Richter, Data curation, Formal analysis, Investigation, Writing – review and editing; Marie-Geneviève Guiraud, Conceptualization, Data curation, Investigation, Methodology,

Writing – review and editing; Mark Roper, Conceptualization, Formal analysis, Validation, Methodology, Writing – review and editing; James AR Marshall, Lars Chittka, Supervision, Funding acquisition, Project administration, Writing – review and editing

**Author ORCIDs**
HaDi MaBouDi https://orcid.org/0000-0002-7612-6465
Jasmin Richter https://orcid.org/0009-0003-9534-7903
Marie-Geneviève Guiraud https://orcid.org/0000-0001-5843-9188
Mark Roper https://orcid.org/0000-0003-1135-6187
James AR Marshall https://orcid.org/0000-0002-1506-167X
Lars Chittka https://orcid.org/0000-0001-8153-1732

**Decision letter and Author response**
Decision letter https://doi.org/10.7554/eLife.106332.sa1
Author response https://doi.org/10.7554/eLife.106332.sa2

## Additional files

**Supplementary files**
MDAR checklist

**Data availability**
The behavioural data and codes for analysing trajectory are available in the public repository figshare at https://doi.org/10.15131/shef.data.14185865.v1.

The following dataset was generated:

| Author(s) | Year | Dataset title | Dataset URL | Database and Identifier |
|---|---|---|---|---|
| MaBouDi HD, Roper M, Guiraud M, Marshall J, Chittka L | 2025 | Data for "Automated video tracking and flight analysis show how bumblebees solve a pattern discrimination task using active vision" | https://doi.org/10.15131/shef.data.14185865.v1 | figshare, 10.15131/shef.data.14185865.v1 |

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
