## [Editor Report]

This useful study revisits a challenging visual learning task in bumblebees. By analyzing the flight trajectory and body orientation in a learning paradigm, the authors discovered an active vision strategy. This contributes to our knowledge of the bumblebee flight behavior. The convincing work will be of interest to researchers working in neuroethology.

---

## [Decision Letter]

**Decision letter after peer review:**

[Editors’ note: the authors submitted for reconsideration following the decision after peer review. What follows is the decision letter after the first round of review.]

Thank you for submitting the paper "Exploring active vision of bees in a simple pattern discrimination task" for consideration by *eLife*. Your article has been reviewed by 3 peer reviewers, and the evaluation has been overseen by a Reviewing Editor and a Senior Editor.

Comments to the Authors:

We are sorry to say that, after consultation with the reviewers, we have decided that this work will not be considered further for publication by *eLife*.

Several aspects of this potentially useful study are novel and interesting. The observation of active vision is exciting and the use of affordable technological solutions was valued by the reviewers. These qualities did however not mitigate the lack of thorough analyses. More details about the paradigm should be provided to ensure that the work is reproducible and more rigorous experiments are necessary to characterize the mechanism of active vision. As a result, the evidence supporting the claims is incomplete and the study is largely inconclusive: it leaves more questions than answers. Prior to the publication of the work, preliminary observations ought to be generalized. At a minimum, the authors should analyze behavior in the transfer test when the animals are either presented with top and bottom half of the x and plus signs.

*Reviewer #1 (Recommendations for the authors):*

The authors present their findings and analysis on the flight behaviour of bumblebees as they perform a visual discrimination task when presented with a rewarding and aversive stimulus. This is a descriptive study where the behaviour of the bees is well explained but I am not sure about the extent of the novelty of the work or the overall impact on the field in general. However, the authors make some interesting observations of the scanning behaviour of the bees and highlight the value of analysing the flight behaviour in these choice experiments. The discussion on nascent and readily available technologies such as cameras on phones and moderate computing power to analyse flight videos to gain deeper insight into animal behaviour is particularly good.

From a neuroethology standpoint, the depth of inference on the bees' visually guided behaviour that the data presented here can provide is difficult to gauge. I say this because while the scanning behaviour is interesting and well described, it is unclear if this is generalizable. For example, there is no data on the scanning behaviour when only the top or bottom half is presented in the transfer test. Do the bees still scan the bottom when only the top half is presented only to find no information and then choose randomly? It is unclear if the scanning behaviour is indeed general. In the same way, if the bees were trained on only the top half and then in transfer presented with the x and plus signs – would they scan only the top section and then would the scanning strategy change? Such questions naturally arise since the data presented in the experiments here cannot clarify if the scanning behaviour observed is indeed general.

If we were to consider approaching the experiment as a "minimization of information redundancy" task where the task is solved by learning the minimum salient information necessary to discriminate the two choices – in this case it would be the part of the pattern in the lower left. The rest of the stimuli is basically redundant information and unnecessary to learn. The bees' behaviour appears to conform to this reasoning, right? If so then another experiment would be needed to support or disprove this and suggest that the scanning strategy is indeed general.

At a minimum it would be good to include analysis of the bees' flight behaviour during the choice in the transfer test when they were either presented with top and bottom half of the x and plus signs.

I would also like the authors to clarify if the bees were able to discriminate if only the top halves of the patterns are presented? I ask because all the figures (Figure 1and 2) indicate that only the top half was presented in the transfer test but then in the text (line 200) it is mentioned that the bees were unable to discriminate for this case and managed to make the correct choice when only the bottom half was presented. If this was indeed the case, then it would be easier to follow if the figures presented the transfer test with the bottom half.

The authors make a statement in the abstract about serial and parallelization and neural processing however it seems like a stretch given the relatively narrow experiment scope performed here. I suggest the authors remove or significantly reduce emphasis to this.

The authors mention flight "dynamics" this should be corrected across the manuscript to flight "trajectories". The former implies analysis of forces which was not performed here.

*Reviewer #2 (Recommendations for the authors):*

This study revisits a challenging visual learning task in bumblebees. By analyzing the flight trajectory and body orientation in a learning paradigm, the authors discovered an active vision strategy.

1) More details about the paradigm should be provided, either in the method section or in the main text. For example, how long does a training bout take on average? How long are two training bouts separated? Were the training bouts and tests conducted within a certain time on the same day? Further, analyses of the learning performance in each training bout and how the behavior changes over training bouts would be highly informative in revealing the task-solving strategies of bees.

2) Although bees' ability to use active vision to solve a task is a fascinating discovery, the study leaves more questions than answers. For example, what would happen if the transfer experiment used only the lower half? What would happen if a single tilted bar or a vertical bar was used? What strategy would bees use if the location of the reward port is placed in other locations of the visual patterns? Does the number of bars in each pattern matter?

This is an exciting discovery and has a huge potential. But the study seemed prematurely wrapped up. More rigorous follow-up experiments are strongly desired.

*Reviewer #3 (Recommendations for the authors):*

This paper largely repeats the findings in Langridge et al. (2021). I am not sure what this paper brings that's new, but maybe I missed something. The paper is difficult to follow, as the structure is not streamlined and the references to figures are chaotic. In addition, large parts of the Methods appear in the Results or figure Legends.

The experiments need to be redone to bring in some novel ideas and to be differentiated from the Langridge study. The paper should also be re-written in a logical and clear manner, that highlights the novelty of the results.

---

## [Author Response]

[Editors’ note: the authors resubmitted a revised version of the paper for consideration. What follows is the authors’ response to the first round of review.]

Several aspects of this potentially useful study are novel and interesting. The observation of active vision is exciting and the use of affordable technological solutions was valued by the reviewers. These qualities did however not mitigate the lack of thorough analyses. More details about the paradigm should be provided to ensure that the work is reproducible and more rigorous experiments are necessary to characterize the mechanism of active vision. As a result, the evidence supporting the claims is incomplete and the study is largely inconclusive: it leaves more questions than answers. Prior to the publication of the work, preliminary observations ought to be generalized. At a minimum, the authors should analyze behavior in the transfer test when the animals are either presented with top and bottom half of the x and plus signs.

These additional experiments have now been performed and new analyses added – see our detailed replies below.

Reviewer #1 (Recommendations for the authors):The authors present their findings and analysis on the flight behaviour of bumblebees as they perform a visual discrimination task when presented with a rewarding and aversive stimulus. This is a descriptive study where the behaviour of the bees is well explained but I am not sure about the extent of the novelty of the work or the overall impact on the field in general. However, the authors make some interesting observations of the scanning behaviour of the bees and highlight the value of analysing the flight behaviour in these choice experiments. The discussion on nascent and readily available technologies such as cameras on phones and moderate computing power to analyse flight videos to gain deeper insight into animal behaviour is particularly good.

Thank you for your thoughtful and detailed feedback. We appreciate your recognition of the value of analysing bumblebee scanning behaviour using camera technologies and the significance of our findings on their visual discrimination strategies. We believe our study provides novel insights into bumblebee visual sampling and decision-making and highlights cost-effective methods for behavioural analysis. We clarified the novelty and impact in the revised manuscript.

From a neuroethology standpoint, the depth of inference on the bees' visually guided behaviour that the data presented here can provide is difficult to gauge. I say this because while the scanning behaviour is interesting and well described, it is unclear if this is generalizable. For example, there is no data on the scanning behaviour when only the top or bottom half is presented in the transfer test. Do the bees still scan the bottom when only the top half is presented only to find no information and then choose randomly? It is unclear if the scanning behaviour is indeed general. In the same way, if the bees were trained on only the top half and then in transfer presented with the x and plus signs – would they scan only the top section and then would the scanning strategy change? Such questions naturally arise since the data presented in the experiments here cannot clarify if the scanning behaviour observed is indeed general.

Thank you for pointing this out. We have conducted a new experiment and performed additional video analysis to address the ‘generalisation’ framework. The updated manuscript has been revised to address your concern. Please see below for more details.

If we were to consider approaching the experiment as a "minimization of information redundancy" task where the task is solved by learning the minimum salient information necessary to discriminate the two choices – in this case it would be the part of the pattern in the lower left. The rest of the stimuli is basically redundant information and unnecessary to learn. The bees' behaviour appears to conform to this reasoning, right? If so then another experiment would be needed to support or disprove this and suggest that the scanning strategy is indeed general.

Thank you for the suggestion. We agree that further experiments were necessary to test whether the scanning strategy reflects a general mechanism for minimising information redundancy. Below, we address your concerns about the novelty, generalisability, and depth of inference provided by our study, with a specific focus on the additional data and analyses included in the revised manuscript.

1. Novelty and Impact of the Study

Our study advances the understanding of bumblebee visual processing by demonstrating their use of an active, feature-based scanning strategy. The novelty lies in showing that bumblebees prioritise specific, task-relevant features of visual stimuli—particularly those in the bottom half of the patterns—rather than uniformly processing the entire pattern. This behaviour reflects a strategy to reduce cognitive load while maximising efficiency in decision-making. The use of bottom-half and top-half transfer tests further underscores this selective strategy, as bees successfully discriminated patterns using cues in the bottom halves but failed to generalise when only the top halves were presented (New figures 2 and 5). This work contributes to the broader field of neuroethology by providing empirical evidence of how bumblebees adapt their scanning behaviours to the most informative regions of their visual environment.

2. Generalisation of Scanning Behaviour

To address your concerns about generalisability, we included additional transfer tests and detailed trajectory analyses. The results show that bees exhibit a strong preference for scanning the lower regions of the stimuli during both training and transfer tests. In the bottom-half test, bees successfully discriminated patterns based on the available cues, demonstrating the robustness of their learned scanning strategy. However, in the top-half test, despite the absence of cues in the lower regions, bees persisted in scanning the bottom half, highlighting the reliance on pre-learned spatial regions for decision-making. These findings suggest that while the scanning strategy is consistent and robust under familiar conditions. It also demonstrates limited flexibility when faced with configurations where the learned visual cues are absent from their previously reinforced regions.

3. Relevance of "Minimisation of Information Redundancy"

While our findings align with aspects of the "minimisation of information redundancy" framework, they also challenge its sufficiency in explaining bumblebee behaviour. The results suggest that bees actively focus on specific, pre-learned features rather than simply reducing redundant information across the entire pattern. The failure to discriminate in the top-half test indicates that their decision-making depends on the availability of key visual features rather than a uniform simplification strategy. Furthermore, analyses of flight trajectories, hovering times, and velocity distributions provide evidence of deliberate and task-specific adjustments in scanning behaviour, reinforcing the role of active feature selection over passive redundancy minimisation.

4. Depth of Inference on Visually Guided Behaviour

Our study provides detailed insights into the mechanisms of bumblebee pattern discrimination through advanced video analyses. Heatmaps and trajectory data reveal targeted scanning and hovering behaviour concentrated on the most informative regions of the patterns. These findings support the hypothesis that bumblebees employ a spatially selective, feature-based strategy to guide their decision-making. While the current experiments focus on specific visual patterns, they establish a foundation for future investigations into the generality and adaptability of these behaviours across different stimuli and environmental conditions.

In summary, we believe that our new data and the additional detailed analyses offer valuable insights into bumblebees' visual scanning behaviour. We sincerely thank you again for your constructive comments, which have significantly enhanced the quality of the manuscript.

At a minimum it would be good to include analysis of the bees' flight behaviour during the choice in the transfer test when they were either presented with top and bottom half of the x and plus signs.

Thank you for your suggestion. We have included an analysis of the bees' flight behaviour during the transfer tests where they were presented with either the top or bottom halves of the stimuli. This analysis provides insights into how bees interacted with these patterns and confirms that their ability to discriminate is primarily driven by the bottom halves of the patterns. The results are now included in the revised manuscript and are supported by updated figures to ensure clarity and consistency.

I would also like the authors to clarify if the bees were able to discriminate if only the top halves of the patterns are presented? I ask because all the figures (Figure 1and 2) indicate that only the top half was presented in the transfer test but then in the text (line 200) it is mentioned that the bees were unable to discriminate for this case and managed to make the correct choice when only the bottom half was presented. If this was indeed the case, then it would be easier to follow if the figures presented the transfer test with the bottom half.

Thank you for pointing this out. We conducted an additional experiment to clarify this question. Consequently, we have refined the text and updated the figures to ensure consistency and provide a clearer representation of the findings.

The authors make a statement in the abstract about serial and parallelization and neural processing however it seems like a stretch given the relatively narrow experiment scope performed here. I suggest the authors remove or significantly reduce emphasis to this.

Thank you for your feedback. We have revised the abstract accordingly.

The authors mention flight "dynamics" this should be corrected across the manuscript to flight "trajectories". The former implies analysis of forces which was not performed here.

The term "flight dynamics" has been replaced with "flight trajectories" throughout the manuscript to ensure accuracy and avoid any misinterpretation

Reviewer #2 (Recommendations for the authors):This study revisits a challenging visual learning task in bumblebees. By analyzing the flight trajectory and body orientation in a learning paradigm, the authors discovered an active vision strategy.

Thank you for highlighting the discovery of an active vision strategy in bumblebees. In the revised manuscript, we have expanded on this finding by including additional analyses of bees' selective scanning behaviours during pattern discrimination. Specifically, we have conducted a new experiment with 20 bees and provided new data from transfer tests, including the top-half test and single-bar orientation tests, which further confirm the role of active vision in bumblebee decision-making. These results strengthen our conclusion that bumblebees actively employ targeted scanning strategies to enhance their visual learning.

1) More details about the paradigm should be provided, either in the method section or in the main text. For example, how long does a training bout take on average? How long are two training bouts separated? Were the training bouts and tests conducted within a certain time on the same day? Further, analyses of the learning performance in each training bout and how the behavior changes over training bouts would be highly informative in revealing the task-solving strategies of bees.

Thank you for the suggestion. The methods section has been improved with additional details about the experimental paradigm. Specifically, we now include information on the duration of training bouts, the intervals between consecutive bouts, and the timing of training and tests within the same day. Additionally, we have analysed the learning performance in each training bout and how behaviour changes across bouts, providing further insights into the task-solving strategies of bumblebees. These updates ensure greater clarity and reproducibility.

2) Although bees' ability to use active vision to solve a task is a fascinating discovery, the study leaves more questions than answers. For example, what would happen if the transfer experiment used only the lower half? What would happen if a single tilted bar or a vertical bar was used? What strategy would bees use if the location of the reward port is placed in other locations of the visual patterns? Does the number of bars in each pattern matter?

Thank you for raising these important points. In the revised manuscript, we have addressed these questions through additional experiments. Specifically:

1. Transfer experiment with the lower half of the patterns: We conducted a transfer test using only the bottom halves of the patterns. The results show that bumblebees were able to successfully discriminate between patterns based on the bottom halves, confirming that this region contains critical visual cues for their decision-making.

2. Transfer experiment with the top half of the patterns: Similarly, we performed a transfer test using only the top halves of the patterns. The results demonstrate that bumblebees could not discriminate between patterns based solely on the top halves, suggesting that they learn to associate the bottom region with relevant visual cues during training.

3. Transfer experiment with single bars: To explore how bees respond to isolated pattern elements, we conducted a single-bar orientation test. The findings reveal that bees exhibit a clear preference for specific bar orientations associated with the rewarded pattern during training.

While we did not explicitly test the impact of changing the reward port's location in this study; our previous observations indicate that relocating the reward port can confuse bees and disrupt their learned associations. We believe these additional experiments and insights provide a clearer understanding of the mechanisms underlying bumblebee visual discrimination and address your concerns effectively.

This is an exciting discovery and has a huge potential. But the study seemed prematurely wrapped up. More rigorous follow-up experiments are strongly desired.

Thank you for highlighting this point. We have revised the text to clarify that the approach paths are consistent for individual chickens but vary across different individuals

Reviewer #3 (Recommendations for the authors):This paper largely repeats the findings in Langridge et al. (2021). I am not sure what this paper brings that's new, but maybe I missed something. The paper is difficult to follow, as the structure is not streamlined and the references to figures are chaotic. In addition, large parts of the Methods appear in the Results or figure Legends.The experiments need to be redone to bring in some novel ideas and to be differentiated from the Langridge study. The paper should also be re-written in a logical and clear manner, that highlights the novelty of the results.

Thank you for your feedback. We have carefully addressed them in the revised manuscript to clarify the unique contributions of our study compared to Langridge et al. (2021). Specifically, our work goes beyond the scope of colour learning to focus on pattern recognition and investigates the role of active vision strategies in bumblebees. This includes advanced analyses of flight paths, body orientations, time hovering and selective scanning behaviours over several tests, offering novel insights into how bumblebees process and distinguish visual patterns. Additionally, to differentiate this work from the Langridge study, we conducted additional experiments, including transfer tests with the top and bottom halves of patterns and single-bar orientation tests. These experiments provide new evidence on how bumblebees use selective scanning strategies to solve visual discrimination tasks, emphasising the novel contributions of this study.

We have also streamlined the manuscript by improving its clarity and organisation. All sections have been restructured to avoid overlap with figure legends, and the overall presentation has been rewritten in a clear and logical sequence to emphasise the novelty and implications of the findings. We believe these changes significantly enhance the manuscript's impact and further differentiate it from prior work.

We hope the following summary highlights the differences and the novelty of our study:

Main Differences Between Langridge et al. (2021) and Our Work:

1. Focus of Study:

Langridge et al. (2021): *Focused on colour learning, investigating how bees use* approach direction prior to landing to inform colour preferences/location during floral pattern recognition.

Current Work: *Focuses on pattern recognition, specifically examining how bumblebees* use active vision strategies to discriminate between geometrical patterns (e.g., a plus sign versus a multiplication sign). Unlike Langridge et al., our study does not involve colour cues but instead highlights shape discrimination.

2. Methodology:

Langridge et al. (2021): Relied on analysing approach direction by extracting limited parameters, such as search time, approach angles, and bee positions. This approach does not provide a detailed understanding of individual bees' inspection strategies or their movements over time.

Current Work: Utilises advanced video analysis tools to track and quantify bee movements, body orientations, hovering time, and scanning behaviours during flight. Our frame-by-frame analysis offers a highly detailed understanding of how bees inspect and interact with visual stimuli.

3. Focus on Active Vision:

Langridge et al. (2021): *While the study indirectly touches on the role of scanning,* it does not explicitly explore active vision or how bees dynamically adjust their visual input. Further experiments would be required to confirm the use of active vision strategies in their setup.

Current Work: *Directly investigates active vision strategies, such as selective scanning* and dynamic visual sampling, which enable bees to identify and focus on specific features of visual patterns. These findings provide novel insights into sensory-motor processes during visual discrimination tasks.

4. Learning Paradigm:

Langridge et al. (2021): *Primarily focused on colour learning, a simpler visual* processing task.

Current Work: *Investigates pattern-based learning and generalisation, exploring how* bees discriminate between geometrical features and transfer learned strategies to novel situations. This represents a significant advancement in understanding bumblebee visual cognition and decision-making.